# Determining the Parameters of the Stinging Nettle (*Urtica dioica* L.) Hydrolate Distillation Process

**DOI:** 10.3390/molecules27123912

**Published:** 2022-06-18

**Authors:** Agnieszka Krajewska, Katarzyna Mietlińska

**Affiliations:** Institute of Natural Products and Cosmetics, Faculty of Biotechnology and Food Sciences, Lodz University of Technology, 90-537 Lodz, Poland; katarzyna.mietlinska@p.lodz.pl

**Keywords:** stinging nettle, *Urtica doica* L., *Urticaceae*, hydrolate, volatile compounds, secondary metabolites

## Abstract

Stinging nettle (*Urtica dioica* L.) is a common perennial herb well known for its therapeutic, cosmetic and food use. Despite the popularity of nettle hydrolate, there is currently no literature describing its composition; likewise, there is still a lack of research describing in detail the parameters of hydrolates in general. *U. dioica* hydrolate fractions were obtained by industrial steam distillation of fresh herb. Total stinging nettle hydrolate was prepared by mixing an equal volume of each fraction. The volatiles were isolated from hydrolate samples by liquid–liquid extraction with diethyl ether, and analysed using GC-FID-MS. Over eighty volatile compounds were identified in *U. dioica* hydrolate. The main group of constituents were oxygenated compounds, mainly alcohols (e.g., (*E*)- and (*Z*)-hex-3-en-1-ol, carvacrol) and oxides (e.g., caryophyllene oxide). The content of volatiles in the representative sample of total hydrolate amounted to 58.2 mg/L. Some qualitative and quantitative changes in the composition of *U. dioica* hydrolate were observed during the progress of distillation. The content of low chain aliphatic alcohols ((*E*)- and (*Z*)-hex-3-en-1-ol) decreased, whereas the percentage of some monoterpene alcohols (carvacrol and α-terpineol) increased. The total content of volatiles in hydrolate also changed and decreased (128.0–6.2 mg/L) during distillation progress. According to our results, to produce stinging nettle hydrolate of good quality, the proper relationship between the amount of hydrolate and raw plant material should result in obtaining 0.74 L hydrolate from 1 kg of fresh stinging nettle herb. Therefore, it may be assumed that the high alcohol content may increase the microbiological stability of the product.

## 1. Introduction

Hydrolates are products of distillation that create pleasant scents and many biological activities. In the past, hydrolates were a waste product of the production of essential oils. Currently, in this age of green chemistry and the trend of zero waste, they are considered a very valuable raw material for many industries, especially perfumery and toiletries, as well as food and cosmetics. More and more often they are also one of the only products of distillation of plant raw materials, but due to plant biodiversity, the production process of hydrolates has not been standardized to date. For this reason, it was considered necessary to determine the hydrodistillation parameters (product volume to column batch ratio) needed to obtain the hydrolate fraction with the highest volatiles content.

Stinging nettle (*Urtica dioica* L., *Urticaceae*) is a plant that is commonly found in Europe, Asia, and Northern America. It grows wild in meadows, backyard gardens, forests, fields and roadside ditches. It is a highly competitive ruderal species that often forms monospecific stands. *U. dioica* is a quite tall (1–2 m), usually dioecious, rhizomatous, perennial herb best known for its stems and leaves being densely covered with stinging hairs, which release potential pain-inducing toxins when contact with them is made. This plant is best known for its medicinal properties [1]. *U. dioica* root and leaf extracts are used as a remedy for prostatic hypertrophy and urinary tract diseases, as well as eczema, menstrual hemorrhage, rheumatism, or anemia [2,3]. Many different classes of organic compounds of medical importance including phytosterols, saponins, flavonoids, tannins, sterols, fatty acids, carotenoids, chlorophylls, proteins, amino acids, and vitamins are produced by stinging nettle [2,3,4]. Dried *U. dioica* leaves are considered a pharmacopoeia raw material [5].

Stinging nettle releases a very low content of volatiles during hydrodistillation, thus it is classified as a non essential oil bearing plant. Nevertheless, some investigations concerning volatile constituents of *U. dioica* herb were recently conducted. According to Ilies et al. [6] hexahydrofarnesylacetone (31.2%), phytol (11.2%), and β-ionone (11.9%) were the main constituents identified in *U. dioica* essential oil. However, essential oil investigated by Gül et al. [7] revealed a different composition. The main constituents were carvacrol (38.2%), naphthalene (8.9%), and carvone (9.0%), although hexahydrofarnesylacetone and phytol were also detected, but in much lower amounts (3.0% and 2.7%, respectively). The unknown composition of stinging nettle’s volatile secondary metabolites, its well-known therapeutic properties, and the availability of this pharmacopoeial material [5] were the reasons that stinging nettle was chosen as the plant material for this study. In the literature, there are currently no studies on the composition, production and use of *U. dioica* hydrolate, therefore, this is the first study showing a composition of the hydrolate obtained industrially from stinging nettle. An additional advantage of this work is the indication of the dependence of the composition of volatile compounds on the volume of fraction collection. 

## 2. Results and Discussion

Seven fractions of stinging nettle hydrolate collected during distillation, and representative total hydrolate prepared by mixing equal volumes of each fraction, were analysed. Every fraction of *U. dioica* hydrolate was colourless liquid with a pleasant, green, herbal scent characteristic of nettle. The pH value (presented in Figure 1) was acidic and ranged from 6.00 to 6.54. The pH values of the analysed samples were quite high in comparison with other hydrolates [8]. The content and composition of volatiles are presented in Table 1. Significant changes in *U. dioica* hydrolate composition during distillation were observed, similarly for other plant materials, e.g., *Rosa rugosa* Thunb [9]. 

The total content of volatiles changed during the progress of the distillation process. At the beginning it increased from 71.2 mg/L in the I fraction to 128.0 mg/L in the III fraction, and then decreased considerably to 49.8 mg/L in the IV and 6.2–10.0 mg/L in other fractions. A similar correlation between the volume of produced hydrolate and the content of volatiles was observed for *R. rugosa* hydrolate [9]. The total content of volatiles in the total hydrolate amounted to 58.2 mg/L and was a little higher than the calculated mean value (52.3 mg/L). It has been reported that the content of organic compounds in hydrolate is around 100–200 mg/L [10]. Conversely, it is obvious that the amount of hydrolate should be in proper relationship to the quantity of plant material. Between 1 and 4 L of hydrolate can be obtained from 1 kg of biomass depending on the particular plant material. However, it is often said that the weight of hydrolate should be equal to that of material [10]. Taking this into account, it could be assumed that hydrolate with quite a high amount of volatiles equal to 86 mg/L could be obtained by mixing the first four fractions. This means that from 1 kg of stinging nettle herb, 0.74 L of hydrolate might be produced.

Over eighty volatile compounds were identified in representative stinging nettle hydrolate. Oxygenated compounds (85.5%) were the principal group of volatile constituents, and among them, alcohols were dominant (73.9%), e.g., (*Z*)-hex-3-en-1-ol (27.9%), carvacrol (10.5%), (*E*)-hex-3-en-1-ol (5.7%), and α-cadinol (4.7%). However, other compounds such as oxides (e.g., *cis*-rose oxide and caryophyllene oxide), ketones (e.g., (*Z*)- and (*E*)-jasmone, verbenone, and car-3-en-2-one), aldehydes (e.g., geranial), esters (e.g., α-terpinyl acetate, ethyl nerolate, bornyl acetate and verbanyl acetate), acids (e.g., citronellic acid, nonanoic acid) were also detected. Some sesquiterpene hydrocarbons, e.g., γ-cadinene (0.4%) and α-calacorene (0.2%), as well as trace amounts of monoterpene hydrocarbons were identified in this product. Total content of hydrocarbons accounted to 0.7%, which corresponded with literature data. Obviously, due to their polarity, oxygenated compounds were the main constituents of every hydrolate [9,10,11,12].

The composition of volatiles in stinging nettle hydrolate changed regularly during the time of hydrodistillation (Figure 2). Short-chain aliphatic alcohols C_6_-C_10_ (mainly (*E*)- and (*Z*)-hex-3-en-1-ol) were the main group of volatiles identified in the I fraction (81.5%). 

Short-chain aliphatic alcohols constituted 22.5% in the II fraction and were scarcely visible in the remaining fractions. On the contrary, the content of monoterpene alcohols (e.g., carvacrol, *p*-cymen-8-ol, *p*-cymen-9-ol, α-terpineol, borneol) revealed a rising tendency. Among sesquiterpene compounds, a slightly different dependence between the content of constituents and volume of produced hydrolate was determined. The content of T-muurolol, α-muurolol, T-cadinol, α-cadinol, spathulenol, and caryophyllene oxide increased in fractions I to III, and then decreased.

Summarizing the obtained results concerning both the total volatiles content and their composition, it was concluded that to produce stinging nettle hydrolate of good quality, the proper relationship between the amount of hydrolate and raw plant material should result in obtaining 0.74 L hydrolate from 1 kg fresh stinging nettle herb. Such products would contain mainly (*E*)- and (*Z*)-hex-3-en-1-ol that are produced in small amounts by most plants [13]. These are important hydrolate ingredients because of their pleasant grassy-green flavour profile. According to Kalemba and Kunicka [14], carvacrol, as well as low chain alcohols, generated the highest microbial activity, therefore, we suspect that the high content of these constituents could positively affect the microbiological stability of hydrolate. Other terpenoid constituents present in hydrolate are also known for their beneficial effect on the human organism [15]. 

It is worth mentioning that the chemical composition of *U. dioica* hydrolate was significantly different to that of previously published research on essential oil. Two available previous reports presented two different essential oils that showed that the profile of stinging nettle volatiles was dependent on the location of growth [6,7]. Understandably, high molecular compounds such as hexahydrofarnesylacetone and phytol, that were present in both previously reported essential oil studies, were not detected in the hydrolate. Nevertheless, some hydrolate volatile compounds identified in this study were previously found in stinging nettle essential oil, e.g., carvacrol was the main constituent in the essential oil of the Turkish herb (32.8%) [7]. The differences between hydrolate volatiles and essential oil composition were likely due to differences in water solubility of oxygenated compounds, especially alcohols and hydrocarbons, than to the location of growth.

## 3. Materials and Methods

### 3.1. Plant Material

Stinging nettle herb (*Urtica dioica* L.) was harvested in May from natural stands in the Northern region of Poland (54°41′59″ N, 18°21′1″ E).

### 3.2. Hydrolate Production

The fresh herb, 14.6 kg, was subjected to two-column distillation apparatus, Innotec-Tetekov TWE 250–2000 VA. During the distillation process that lasted 100 min, seven hydrolate fractions were collected (2.7 L each, 18.9 L in total). A representative sample of hydrolate called “total hydrolate” was also prepared by mixing an equal volume (250 mL) of each fraction. The essential oil was not obtained during the distillation process.

### 3.3. pH Value

The pH value of every hydrolate fraction and total hydrolate was determined using a CP-511 pH meter with IJ44A electrode (Elmetron Company, Zabrze, Poland). The measurements were performed at 20 °C.

### 3.4. Isolation of Volatile Compounds from Hydrolate

Volatile compounds were isolated from each hydrolate fraction and total hydrolate by liquid–liquid extraction with diethyl ether. The hydrolate sample (500 mL) was salted with NaCl (180 g) to reduce the solubility of the volatile compounds in water, and extracted with 100 mL of solvent. The extraction process was repeated five times, each time using a fresh amount of diethyl ether. Extracts were merged to obtain a final 500 mL of diethyl ether extract from each fraction of the hydrolate (I–VII) and total hydrolate. Extracts were dried over anhydrous sodium sulfate and filtered. The solvent was removed using a rotary vacuum evaporator at 36 °C and under the pressure of 25 mmHg. The remaining mixture of volatiles was weighed and the content of volatiles was reported as mg/L. The composition of isolated volatiles was analysed by GC-FID-MS method.

### 3.5. Analysis of Volatile Compounds

Volatile compounds isolated from hydrolate fractions were analysed by gas chromatography coupled with mass spectrometry (GC-FID-MS).

Apparatus: Trace GC Ultra gas chromatograph coupled with DSQ II mass spectrometer (Thermo Electron Corporation), non-polar capillary column Rtx-1 ms (60 m × 0.25 mm, 0.25 m film thickness), programmed temperature: 50 (3 min)–300 °C, 4 °C/min, injector (SSL) temperature 280 °C, detector (FID) temperature 300 °C, transfer line temperature 250 °C, carrier gas—helium, flow with constant pressure 200 kPa, split ratio 1:20. The mass spectrometer parameters: ion source temperature 200 °C, ionization energy 70 eV (EI), scan mode: full scan, mass range 33–420. The percentages of constituents were computed from the GC peak area without using a correction factor.

Identification of the components was based on a comparison of their mass spectra with literature data as well as linear retention indices (RI, non-polar column), determined with reference to a series of n-alkanes C_8_–C_24_, by comparing with those in Adams [16] as well as in computer libraries: NIST 2011 (USA) and MassFinder 4.1 (Germany).

## 4. Conclusions

Over eighty types of volatiles were identified in stinging nettle hydrolate. The main group of constituents was alcohols: (*Z*)- and (*E*)-hex-3-en-1-ol, carvacrol, α-cadinol (4.7%), *p*-cymen-9-ol, and T-muurolol. The content of total volatiles and their quantitative composition was related to the volume of produced hydrolate and changed with the following fractions. Some regular changes in the first four fractions (I–IV) were observed. The content of short-chain aliphatic alcohols decreased, while the content of mono- and sesquiterpene alcohols increased.

According to the content and chemical composition of volatiles, the first four fractions (I–IV) of stinging nettle hydrolate were considered as the most valuable. Due to this statement, to obtain a product with good quality, from 1 kg of fresh plant material, 0.74 L of *U. dioica* hydrolate might be produced. It is worth noting that this is one of the few studies showing the dependence of the content of volatile components on the amount of the obtained hydrolate, and the first one to our knowledge, showing the qualitative and quantitative composition of the volatile components of the stinging nettle hydrolate. Continuous interest in using this raw material, especially in the food and cosmetic industries, indicates the need for further research in this field.

## Figures and Tables

**Figure 1 molecules-27-03912-f001:**
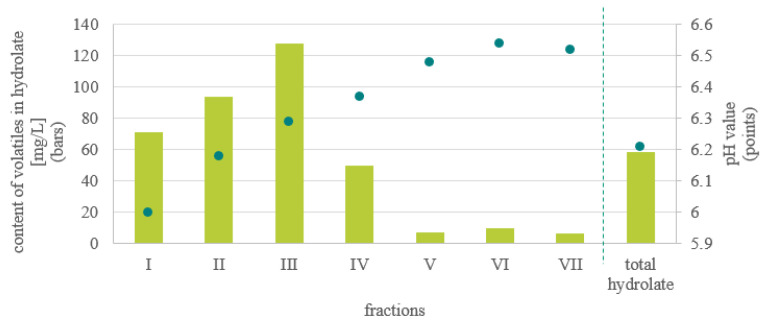
pH value (green bullet line) and content of volatile compounds (green bars) in hydrolate fractions and total hydrolate of the *U. dioica* L.

**Figure 2 molecules-27-03912-f002:**
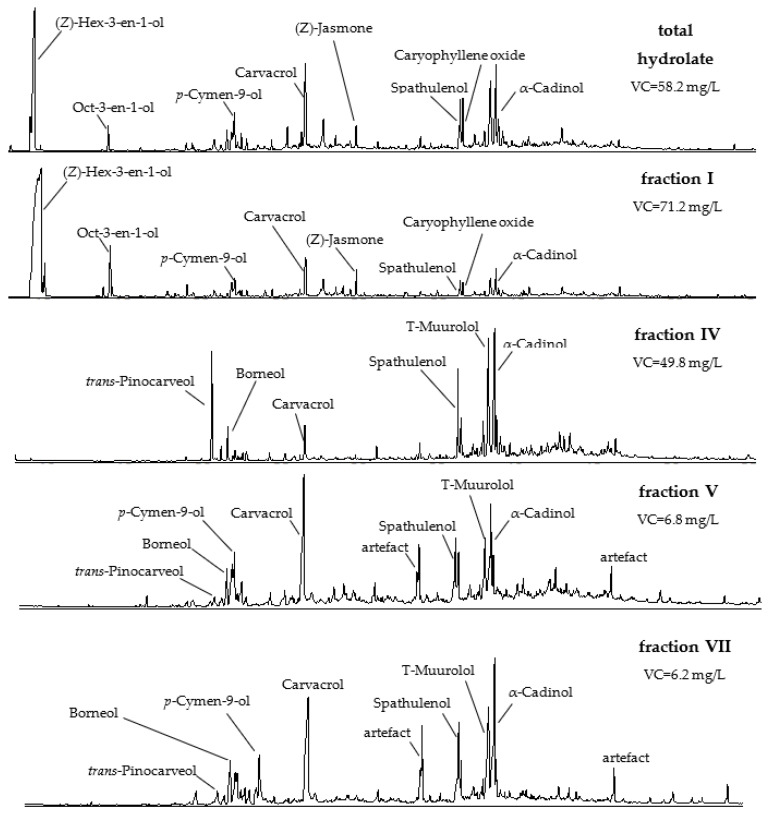
Changes in the profile of the *U. dioica* L. hydrolate volatile compounds, in total and by fraction (I, IV, V and VII).

**Table 1 molecules-27-03912-t001:** Composition of stinging nettle (*Urtica dioica* L.) hydrolate.

No	Compound	RI_lit_	RI_exp_	Total Hydrolate	Hydrolate Fraction						
					I	II	III	IV	V	VI	VII
				[%]							
1.	(*E*)-**Hex-3-en-1-ol**	836	833	**5.7**	**0.5**	**7.1**	**-**	**-**	**-**	**0.2**	**-**
2.	(*Z*)**-Hex-3-en-1-ol**	841	841	**27.9**	**73.7**	**14.3**	**-**	**-**	**-**	**-**	**-**
3.	Hexan-1-ol	854	851	0.5	1.8	0.1	-	-	-	-	-
4.	(*Z*)-Hex-2-en-1-ol	951	857	t	0.4	t	-	-	-	-	t
5.	α-Thujene	926	920	-	-	-	-	t	t	-	0.1
6.	α-Pinene	934	928	-	-	-	-	t	t	-	t
7.	Camphene	950	944	-	-	-	-	t	t	-	-
8.	6-Methyl-2-heptanol	950	952	0.1	0.1	0.4	-	-	-	-	-
9.	Heptan-1-ol	957	954	-	0.5	-	-	-	-	-	-
10.	Oct-1-en-3-ol	962	962	1.6	3.4	0.5	-	-	-	-	-
11.	Allyl isovalerate	920 *^a^*	966	0.2	0.7	-	-	-	-	-	-
12.	Octan-4-ol	975	974	t	0.1	-	-	-	-	-	-
13.	Dehydro-1,8-cineol	979	974	-	-	-	-	-	-	0.4	-
14.	2,6-Dimethylhept-6-en-2-ol *	- *^b^*	1009	0.1	t	-	-	-	-	-	-
15.	*p*-Cymene	1014	1010	t	t	0.1	-	t	t	-	t
16.	Limonene	1025	1017	t	0.1	t	-	t	0.8	-	t
17.	(*Z*)-Oct-5-en-1-ol	1048	1052	t	0.1	-	-	-	-	-	-
18.	Octan-1-ol	1057	1053	t	0.3	0.1	-	-	-	-	-
19.	*trans*-Linalool oxide (f)	1058	1056	t	0.1	t	t	t	t	0.2	t
20.	Heptanoic acid	1068	1068	t	0.1	-	-	-	-	-	-
21.	*cis*-Linalool oxide (f)	1072	1071	t	0.2	t	-	t	t	-	-
22.	Linalool	1086	1085	0.2	0.2	0.1	0.1	0.4	0.4	1.0	0.2
23.	β-Phenyloethanol	1088	1085	0.2	0.2	0.4	-	t	-	t	-
24.	Benzeneacetonitrile	1089	1092	0.1	0.1	-	-	-	0.3	-	-
25.	*cis*-Rose oxide	1096	1096	0.1	0.1	-	-	-	-	0.8	-
26.	*trans*-Rose oxide	1116	1121	t	t	0.1	-	t	0.2	-	-
27.	*trans*-Pinocarveol	1126	1123	0.1	t	0.1	-	0.1	0.2	-	-
28.	*trans*-Verbenol	1134	1129	0.7	0.3	0.6	-	t	0.7	t	1.6
29.	Isopulegol	1135	1130	0.6	0.1	0.6	-	-	-	3.2	-
30.	Citronellal	1135	1130	t	t	t	-	-	t	-	-
31.	Camphene hydrate	1137	1131	-	-	-	-	0.2	-	t	t
32.	Pinocarvone	1144	1138	t	0.2	-	-	1.6	t	-	-
33.	*p*-Mentha-1,5-dien-8-ol	1148	1140	0.4	t	t	-	-	0.8	-	1.0
34.	Isomenthone	1146	1141	-	-	-	-	t	0.2	-	-
35.	Isoborneol	1148	1142	-	-	-	-	t	2.0	-	-
36.	Isoisopulegol	1150	1143	t	t	0.1	-		t	-	-
37.	Borneol	1155	1149	1.1	0.1	0.4	0.5	3.0	2.0	0.7	4.5
38.	Nonan-1-ol	1163	1157	t	0.6	-	-	-		-	-
39.	*p*-Cymen-8-ol	1158	1158	1.6	0.2	2.0	1.6	0.5	5.4	3.3	3.4
40.	*trans*-Car-2-en-4-ol	- *^b^*	1159	t	0.2	0.9	1.5	0.1	-	-	1.7
41.	*p* **-Cymen-9-ol**	1168	1162	**3.2**	**0.7**	**3.4**	**3.2**	**1.4**	**6.5**	**3.2**	**1.7**
42.	2-Methylisoborneol	1164	1167	0.3	0.1	0.7	1.0	0.3	0.3	t	0.5
43.	α-Terpineol	1176	1173	0.9	0.2	0.9	1.4	0.6	2.4	1.7	1.3
44.	Myrtenal	1174	1174	t	t	t	t	0.2	t	t	t
45.	Verbenone	1183	1181	0.7	0.2	1.3	1.4	t	0.8	0.3	1.5
46.	Myrtenol	1184	1184	t	t	t	t	1.4	t	t	t
47.	Coumaran	1191	1198	0.4	0.5	t	0.2	t	0.1	2.3	8.2
48.	Citronellol	1210	1211	0.2	0.3	0.2	1.7	-	t	-	-
49.	Neral	1218	1214	-	-	-	1.2	-	-	-	-
50.	Carvone	1218	1215	t	-	-	-	t	0.2	-	0.1
51.	2,2-Dimethyloct-4-enal *	- *^b^*	1216	-	-	-	-	0.7	-	-	-
52.	Car-3-en-2-one	1245	1222	0.7	-	1.0	1.1	t	1.3	-	0.6
53.	Piperitone	1232	1227	t	t	-	-	-	-	0.3	-
54.	Geraniol	1238	1237	t	0.1	-	-	-	-	-	-
55.	Geranial	1247	1242	0.2	t	-	2.5	t	-	-	0.3
56.	*trans*-Ascaridol glycol	1266 *^a^*	1244	0.1	0.1	0.7	-	0.6	1.6	-	0.1
57.	*p*-Cymen-7-ol	1269	1264	-	-	0.5	-	t	t	-	t
58.	Nonanoic acid	1262	1264	0.1	0.1	-	-	-	-	-	-
59.	Bornyl acetate	1273	1269	0.2	0.2	0.6	0.5	0.3	0.1	t	t
60.	**Carvacrol**	1282	1280	**10.5**	**2.5**	**16.9**	**22.0**	**4.5**	**22.0**	**10.4**	**23.4**
61.	Citronellic acid	1300	1302	0.1	t	0.3	-	t	-	-	-
62.	**8-Hydroxyneomentol**	1310	1310	**2.7**	**0.2**	**1.2**	**0.4**	**0.4**	**0.2**	**0.4**	**0.2**
63.	8-Hydroxymenthol (isomer) *	- *^b^*	1325	0.1	0.3	0.1	-	0.1	t	0.4	t
64.	8-Hydroxymenthol (izomer) *	- *^b^*	1331	0.7	-	-	-	0.1	-	2.1	t
65.	α-Terpinyl acetate	1334	1332	0.5	0.1	0.4	0.4	0.1	0.8	t	t
66.	Ethyl nerolate	1335	1338	0.3	0.1	-	-	t	-	0.3	-
67.	5,9-Dimethyldeca-4,8-dienal *	- *^b^*	1339	-	0.2	-	-	-	-	-	-
68.	(*E*)-Jasmone	1359	1358	0.3	0.2	1.6	t	0.1	t	t	-
69.	(*Z*)-Jasmone	1369	1367	1.3	1.0	1.2	0.3	0.3	0.9	0.2	0.3
70.	Methyleugenol	1375	1370	0.1	t	t	t	t	0.3	0.1	0.1
71.	*p*-Menth-2-ene-1,7-diol monoacetate	- *^b^*	1407	0.4	t	0.6	0.8	1.0	1.1	0.2	1.0
72.	β-Ionone epoxide	1460	1461	0.1	0.2	0.1	0.1	0.1	t	0.1	-
73.	Dihydroactinidiolide	1495	1489	0.8	0.2	1.4	1.6	1.6	2.0	2.3	2.4
74.	γ-Cadinene	1512	1508	0.4	0.1	0.4	0.2	0.7	t	t	0.4
75.	α-Calacorene	1534	1528	0.2	t	0.2	0.2	0.3	0.4	0.2	0.3
76.	β-Caryophyllene oxide	1546	1541	0.1	t	0.1	t	0.2	0.1	0.5	t
77.	γ-Calacorene	1551	1547	t	t	t	t	t	t	t	t
78.	**Spathulenol**	1569	1566	**2.8**	**0.5**	**3.8**	**7.6**	**8.2**	**4.8**	**7.2**	**5.9**
79.	**Caryophylene oxide**	1578	1572	**3.1**	**0.4**	**2.8**	**3.6**	**3.9**	**3.6**	**8.2**	**2.6**
80.	Aromadendrene epoxide	1590	1592	0.1	-	-	-	-	-	-	-
81.	6-*epi*-Cubenol	1602	1603	0.2	t	0.2	1.0	t	0.1	t	0.2
82.	1,10-*diepi*-Cubenol	1615	1616	0.7	0.1	0.8	2.0	2.4	0.6	0.3	0.8
83.	**T-Muurolol**	1628	1626	**2.4**	**0.4**	**2.6**	**6.4**	**7.1**	**3.4**	**4.0**	**4.5**
84.	**α-Muurolol**	1628	1628	**1.2**	**0.2**	**1.4**	**3.5**	**3.6**	**0.9**	**2.1**	**4.2**
85.	**T-Cadinol**	1633	1629	**1.1**	**0.2**	**1.3**	**3.2**	**3.6**	**1.1**	**1.3**	**t**
86.	**α-Cadinol**	1643	1640	**4.7**	**0.7**	**5.6**	**12.7**	**13.0**	**6.8**	**3.9**	**3.5**
87.	*trans*-Calamen-10-ol	1669 *^a^*	1645	1.1	0.2	1.2	1.4	2.9	1.8	0.6	8.7
88.	Eudesma-4(15),7-dien-1β-ol	1688 *^a^*	1661	t	0.1	0.2	t	0.9	0.2	0.3	t
89.	*cis*-14-*nor*-Muurol-5-en-4-one	1684 *^a^*	1664	0.2	t	0.3	0.3	0.7	0.1	0.1	0.2
90.	10-*nor*-Calamen-10-one	1684	1673	0.3	t	0.4	0.5	1.1	0.4	0.1	0.3
91.	Oplopanone	1715	1710	0.6	0.3	0.7	0.7	0.7	0.6	0.3	0.3
92.	Oplopanonyl acetate	1791	1791	0.1	-	-	-	1.2	-	0.2	-
93.	Isopropyl tetradecanoate	1827 *^a^*	1808	0.2	t	0.6	0.6	1.3	0.4	0.2	0.4
94.	Platambin	1867 *^a^*	1833	0.4	0.1	0.3	t	0.6	0.3	0.1	0.2
**Total content of oxygenated compounds**				**85.5**	**94.9**	**83.2**	**87.2**	**71.2**	**78.1**	**65.9**	**86.4**
Alcohols:				73.9	89.6	69.3	71.2	55.9	64.5	48.7	67.6
Aliphatic alcohols				35.9	81.5	22.5	0.0	0.0	0.0	2.5	0.0
Monoterpene alcohols				23.4	5.6	29.4	33.4	13.6	44.5	26.4	39.6
Sesquiterpene alcohols				14.6	2.5	17.4	37.8	42.3	20.0	19.8	28.0
**Hydrocarbons**				**0.7**	**0.3**	**0.7**	**0.4**	**1.0**	**1.5**	**0.2**	**0.8**
**Total indentified [%]**				**86.2**	**95.2**	**83.9**	**87.6**	**72.2**	**79.6**	**66.1**	**87.2**
**Content of volatiles in hydrolate (VC) [mg/L]**				**58.2**	**71.2**	**94.0**	**128.0**	**49.8**	**6.8**	**10.0**	**6.2**

RI_lit_, literature retention index; RI_exp_, experimental retention index; t, trace (<0.05%). * isomer not identified. *^a^* RI on column of other polarity. *^b^* RI not available.

## Data Availability

The data presented in this study are available on request from the corresponding author.

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
