# Peer review of "Determining the Parameters of the Stinging Nettle (Urtica dioica L.) Hydrolate Distillation Process"

_molecules, 2022, doi:10.3390/molecules27123912_

Round 1

Reviewer 1 Report

The manuscript represents an investigation on the stinging nettle hydrolate. The study is performed with the aim to elucidate the qualitative and quantitative composition of the hydrolate. This data is provided for the first time and represents a new information.

The product of a plant hydro-distillation is a heterogeneous mixture containing small drops of essential oil dispersed in the hydrolate. In an ideal case, after a full separation of the essential oil, the hydrolate is a homogeneous liquid fraction, which contains only water-soluble substances. In practice, a hydrolate might contain invisible microscopic drops of unseparated essential oil. In the case of nettle, which is considered as a low-essential-oil plant, the production of hydrolate only is justified.

Analyses are made of a number of consecutive fractions of hydrolate and of the total hydrolate. This approach makes possible to follow the composition changes in the course of time. At this base, the conditions for production of hydrolate with richest composition are obtained.

The conclusions for composition and concentration changes are logical, because more volatile compounds are distilled first, followed by less volatiles. In the course of time the raw material is exhausted, so, it is not surprising that the concentration of bio-compounds in the hydrolate becomes lesser. A positive quality of the obtained hydrolate is its microbiological stability due to high alcohol content.

Modern analytical technique is used for identification of the substances, and the results are trustful.

The manuscript represents new scientific information about the chemical composition of nettle hydrolate. It also supplies useful practical information for organizing the hydro-distillation of nettle with the aim of hydrolate production.

I recommend the acceptance of this manuscript in the present form.

Author Response

Dear Reviewer,

Regarding to your revision of my manuscript molecules-1782460 I would like to thank you for your kind remarks.

Best regards.

Reviewer 2 Report

The manuscript titled: Determining the parameters of the stinging nettle (Urtica dio- 2 ica L.) hydrolate distillation process, describes important process which can benefit industry in addition to other pharmaceutical sector, yet few issues should be addressed before further step:

Abstract

It is advised to make the final part of the abstract look more conclusive

Results

-Line 118: explain this sentence more : (Moreover, the high content of these alcohols as well as car- 118 vacrol could positively affect the microbiological stability of hydrolate)

-Line 129: it is advised to make a chart or drawing to show the conclusion/ finding of this study, bearing in mind the final statement: (The differences 129 between hydrolate volatiles and essential oil composition are rather due to differences in 130 water solubility of oxygenated compounds, especially alcohols and hydrocarbons than to 131 growing place).

-It is generally advised to write (first one, up to our knowledge, showing….) rather than writing: (first one showing…).

Author Response

Dear Reviewer,

Regarding to your revision of my manuscript molecules-1782460 I would like to thank you for your kind remarks. I would like also answer your question and explain some inconsistencies.

  1. Abstract - It is advised to make the final part of the abstract look more conclusive.

It was corrected to: According to our results to produce stinging nettle hydrolate of good quality the proper relationship between the amount of hydrolate and raw plant material should have amounted to 0.74 L hydrolate from 1 kg of fresh stinging nettle herb. Therefore it may be assumed that the high alcohol content may increase the microbiological stability of the product.

  1. -Line 118: explain this sentence more : (Moreover, the high content of these alcohols as well as carvacrol could positively affect the microbiological stability of hydrolate)

It was corrected to: According to Kalemba and Kunicka [14] carvacrol as well as low chain alcohols revealed hight microbial activity therefore we suspect that, the high content of these constituents could positively affect the microbial stability of hydrolate. (…)

[14] Kalemba, D.; Kunicka, A. Antibacterial and Antifungal Properties of Essential Oils. Curr. Med. Chem. 2003, 10, 813–829, doi:10.2174/0929867033457719.

  1. Line 129: it is advised to make a chart or drawing to show the conclusion/ finding of this study, bearing in mind the final statement: (The differences between hydrolate volatiles and essential oil composition are rather due to differences in water solubility of oxygenated compounds, especially alcohols and hydrocarbons than to growing place).

The described research concerned only the production of nettle hydrolate. The essential oil was not analyzed. Data concerning oil come from the literature. Therefore, the preparation of a chart is not possible in this manuscript. Thank you for the suggestion, I will prepare such chart in my next manuscript concerning production of essential oil and hydrolate from other plant material.

  1. -It is generally advised to write (first one, up to our knowledge, showing….) rather than writing: (first one showing…). – It was corrected in the manuscript.

Best regards.